# Identification of Antimicrobial Compounds from *Sandwithia guyanensis*-Associated Endophyte Using Molecular Network Approach

**DOI:** 10.3390/plants9010047

**Published:** 2019-12-29

**Authors:** Phuong-Y Mai, Marceau Levasseur, Didier Buisson, David Touboul, Véronique Eparvier

**Affiliations:** 1Paris-Saclay CNRS ICSN, Institut de Chimie des Substances Naturelles UPR 2301, Université Paris Saclay, 91198 Gif-sur-Yvette, Francemarceau.levasseur@cnrs.fr (M.L.); david.touboul@cnrs.fr (D.T.); 2Museum National d’Histoire Naturelle, Molécules de Communication et Adaptation des Micro-organismes, UMR 7245 CNRS/MNHN, Sorbonne Université, Paris CEDEX 05, 75006 Paris, France; didier.buisson@mnhn.fr

**Keywords:** endophytes, latex, Molecular Networking, antimicrobial, stephensiolides, *Lecanicillium* sp.

## Abstract

The emergence of multidrug resistant bacterial pathogens and the increase of antimicrobial resistance constitutes a major health challenge, leading to intense research efforts being focused on the discovery of novel antimicrobial compounds. In this study, endophytes were isolated from different parts of *Sandwithia guyanensis* plant (leaves, wood and latex) belonging to the Euphorbiaceae family and known to produce antimicrobial compounds, and chemically characterised using Molecular Network in order to discover novel antimicrobial molecules. One fungal endophyte extract obtained from *S. guyanensis* latex showed significant antimicrobial activity with Minimal Inhibitory Concentration on methicillin-resistant *Staphylococcus aureus* at 16 µg/mL. The chemical investigation of this fungus (*Lecanicillium* genus) extract led to the isolation of 5 stephensiolides compounds, four of which demonstrated antibacterial activity. Stephensiolide I and G showed the highest antibacterial activity on MRSA with a MIC at 4 and 16 µg/mL respectively.

## 1. Introduction

The search for antimicrobial molecules represents a global challenge in the field of public health. Indeed, antimicrobial resistance threatens the very core of modern medicine for infectious diseases. Systematic misuse and overuse of drugs in human medicine and food production have put every nation at risk. Few replacement products are in the pipeline. Without harmonised and immediate action on a global scale, the world is heading towards a post-antibiotic era in which common infections could once again kill [1,2]. Therefore, the search for new antimicrobial compounds amidst the rising antibiotic resistance is an urgent need and a major challenge for scientific community and public health. Over the last two decades, although the number of new compounds being isolated each year in natural products has been constant, the discovery rate of new antimicrobial compounds has declined [3]. Several metabolomic profiling methodologies have recently been implemented to minimise the high rediscovery rate associated with natural product research and restrict the high amounts of data generated by the analysis of these complex extracts [4]. Molecular networking (MN) approaches allow the organisation of untargeted tandem Mass Spectrometry (MS) datasets according to their spectral similarity and generate clusters of structurally related metabolites. These approaches offer powerful tools for navigating the chemical space of complex biological systems and can be used to view the chemical constituents of a wide variety of extracts in a single map [5,6]. Furthermore, MN offers the possibility to map additional information, such as biological, analytical and taxonomical details over networks [6,7,8]. In this study, we applied a related multi-informative prioritisation strategy based on the use of MN, with the aim of finding new antimicrobial structures from of a set of extracts from 42 Euphorbiaceae endophytes.

The Euphorbiaceae family comprises approximately 283 genera and 7300 species, and has been found in various morphological types, including herbs, shrubs and trees [9]. Some species of the family are used as lubricants and purgatives, while others are used as a source of medicinal drugs for the treatment of diarrhoea, inflammation and swelling. Their milky latex secretions impart several toxic effects such as dermatitis, severe irritation to the eyes and stomach discomfort. Moreover, *Euphorbia* sp. latex has been shown to possess moderate antimicrobial properties and ability to protect against herbivores [10]. Together, the antimicrobial, insecticidal and toxicity properties of *Euphorbia* latex makes it a hostile environment for microbes and plant-feeding insect pests. However, it has been recently highlighted that *Euphorbia* sp. latex contains unexpectedly complex bacterial and fungal communities [11]. Meanwhile, endophytes, which are microorganisms residing in plant tissues without inducing any diseases in the host plants, have been shown to defend themselves as well the plants from pathogens and have a crucial role in plant survival. They also improve the ecological adaptability of the hosts by enhancing their tolerance to environmental stresses and resistance to phytopathogens and/or herbivores [12]. In addition, endophytes residing within plants may produce identical or similar secondary metabolites as their host plant, and therefore, have also been recognised as a potential source of novel bioactive molecules. These common compounds are produced as a result of bidirectional horizontal gene transfer between the host plant and the associated endophytes. The endophytes are also known to produce enzymes involved in the biosynthetic pathway of plant metabolites. Further, natural endophyte–host plant interaction stimulates the production of secondary metabolites by the host plant [13]. Several reports suggest that endophytes produce various bioactive compounds that are identified in endophytes and host plants [14,15]. 

This study focused on hitherto unexplored wood and latex endophytes isolated from *Sandwithia guyanensis* plant, an endemic species of the Guiana shield, which was phytochemically investigated in a previous work [16]. Moreover, 42 cultivable endophytic strains were isolated from different parts of 3 individual parts of the *S. guyanensis* plant (from wood (4), latex (14) and leaves (24)). This collection was cultured, extracted and screened against human pathogens (ATTC and clinical strains). Additionally, the full strain collection was analysed by high-resolution tandem mass spectrometry profiling (HRMS/MS), and the resulting fragmentation data were organised as molecular networks (MNs) for rapid and efficient de-replication. 

## 2. Results

### 2.1. Biological Activities of Endophytic Extracts

The 42 crude extracts obtained from the isolated strains were biologically tested (Appendix A). A growth inhibition was observed on methicillin-resistant *Staphylococcus aureus* (MRSA) for 9 of them, 5 inhibited *Candida albicans* and 7 acted on *Trichophyton rubrum* (MIC ≤ 64 µg/mL). Amongst those, 2 extracts strongly inhibited the growth of *C. albicans* at MIC ≤ 8 µg/mL and 6 extracts presented a strong antifungal activity on *T. rubrum* at MIC ≤ 8 µg/mL. With this analysis, the 13 active extracts (Appendix A) were tested for cytotoxicity on MRC5 cell line. At 10 µg/mL, 4 extracts presented more than 50 % inhibition of cell proliferation and viability, and at 1 µg/mL, only one extract was found to confer cytoxicity. Among the 14 extracts isolated from latex, 5 exhibited an MIC ≤ 64 µg/mL on MRSA (Figure 1), with the most active extract on MRSA showing an MIC ≤ 16 µg/mL (BSNB-SG3.7) without potential cytotoxicity.

### 2.2. Metabolomic Analyses

All active and inactive ethyl acetate (EtOAc) extracts were analysed (corresponding to the metabolome of 42 endophytic strains) by Ultra-High-Performance Liquid Chromatography-High-Resolution Mass Spectrometry (UHPLC-HRMS/MS) using the data-dependent acquisition mode. Data were first processed by MZmine 2.38 [17,18], and molecular networks (MNs) were then calculated and visualised using MetGem version 1.2.0 RC2. Relative quantification of the ions based on the respective areas of the corresponding extracted ion chromatograph (XIC) was represented as pie chart diagrams, with their proportions. Molecular networks were then mapped with taxonomic information: one colour per strain in the pie charts (Appendix A); with biological activities on MRSA (Figure 2A), on *C. albicans* (Appendix A) and on *T. rubrum* (Appendix A), respectively; and with plant-part origin, i.e., yellow colour for strains isolated from latex (Appendix A). An initial MN was built with the data from the 42 extracts (Figure 2) composed of 3393 nodes. BSNB-SG3.7 strain extract showed the best activity against MRSA with least cytotoxicity and an ability to produce unique molecules (Pink clusters surrounded by red – Figure 2A), and was then chosen for further investigation. Standard and analogues of the molecules in this cluster has been sought, they were matched their MS^2^ spectra with all the referenced molecules available in the libraries (Figure 2B). 

### 2.3. Isolation and Identification of Molecules from Lecanicillium sp. (BSNB-SG3.7 Strain)

The EtOAc extract of BSNB-SG3.7 strain, identified as *Lecanicillium* sp., was subjected to preparative liquid chromatography (HPLC) using a C_18_ silica gel column to yield 5 pure compounds. 

Compound **1** showed a molecular ion peak [M+H]^+^ at *m/z* 670.4401 (calculated for 670.4386, Δm = +2.2 ppm) in the ESI-HRMS spectrum, consistent with the molecular formula of C_33_H_59_N_5_O_9_. Analysis of 1D and 2D NMR spectra of compound **1** indicated the presence of five amino acids: threonine, valine, isoleucine and two serine residues and the presence of long alkyl chain, leading to its identification as a stephensiolide I analogue upon comparison with existing literature (Figure 3) [19,20]. The MS/MS spectrum of compound **1** confirmed the previous annotated structure through the detection of immonium ions and the classical peptide sequencing. For example, the difference between ion peaks at *m/z* 670.4380 and *m/z* 539.3401 corresponded to the mass increment of isoleucine and one water molecule. The number of carbon and hydrogen atoms of the acyl chain could be also determined according to the detection of the ion peak at *m/z* 183.1706 (Appendix A). 

Thereafter, the ESI-HRMS of compound **2** showed a molecular ion peak [M+H]^+^ at *m/z* 642.4050 (calcd for 642.4073, Δm = −3.6 ppm), consistent with the molecular formula of C_31_H_55_N_5_O_9_. This compound showed a 1D NMR spectrum similar to compound **1** indicating highly similar structures. The difference of 28 Da observed between the two molecular masses correspond to the loss of 2 -CH_2_ groups from the acyl chain from compound **1**. This structure of compound **2** was confirmed by the MS/MS spectrum showing an ion peak at *m/z* 155.1437 (Appendix A). The ESI-HRMS of compound **3** showed a molecular ion peak [M+H]^+^ at *m/z* 656.4243, consistent with the molecular formula of C_32_H_57_N_5_O_9_ (calcd for 656.4229, Δm = +2.1 ppm). The difference of 14 Da between compounds **1** and **3** correspond to the loss of a -CH_2_ group that can be attributed to the replacement of an isoleucine with valine. The protons of valine and isoleucine from compound **1** at position (δ_C_ 58.9; δ_H_ 4.14) and (δ_C_ 60.1; δ_H_ 4.23), respectively, were found at the same chemical shift in the 1D NMR spectrum of compound **3** and are integrated for 2H (δ_C_ 59.6; δ_H_ 4.19), supporting the presence of two valine in its structure. Moreover, none of the ion molecular peaks corresponded to isoleucine on its MS/MS spectrum, and the relative intensity of the ion peak corresponding to valine was found to be markedly higher than the other ion peaks, thereby confirming the structure of compound **3** (Appendix A).

The ESI-HRMS of compound **4** showed a molecular ion peak [M+H]^+^ at *m/z* 628.3936 (calcd for 628.3916, Δm = +3.1 ppm), consistent with the molecular formula of C_30_H_53_N_5_O_9_. The difference of 42 Da between compounds **1** and **4** correspond to the loss of 3 -CH_2_ groups with 2 -CH_2_ groups from the acyl chain and 1 -CH_2_ group from an amino acid. The ^1^H NMR peaks at position 4.18 ppm integrate for 2H (Appendix A) and have the same profile as for compound **3**, indicating that isoleucine from compound **3** was replaced by valine for compound **4**. The longer length of the alkyl chain was confirmed by the MS/MS spectrum with an ion peak at *m/z* 155.000 as compound **2** (Appendix A). Finally, the ESI-HRMS of compound **5** showed a molecular ion peak [M+H]^+^ at *m/z* 668.4201 (calcd for 668.4229, Δm = −4.1 ppm), consistent with the molecular formula of C_33_H_57_N_5_O_9_ (Appendix A). The difference of 2 Da observed between compounds **1** and **5** indicate the loss of 2 protons, corresponding to a double bond in the acyl chain. This hypothesis was confirmed according to the protons at 5.42 ppm integrating for 3H (1 proton attached to CH_3_ and 2 protons directly attached to a C=C), and by an ion peak at *m/z* 181.1596 observed in MS/MS mass spectrum (Appendix A). The HMBC correlations ^1^H-^13^C between protons 1 with C-2 and C-3 as well as the correlations of H-5 and H-6 with C-3 and C-4 permitted to fix the double bond at position at C-5–C-6 of the acyl chain (Appendix A). Unfortunately, the coupling constants are not observable for the double bond protons and therefore, the configuration of the latter could not be determined [19].

The isolated compounds were identified as stephensiolides I (**1**), D (**2**), G (**3**), C (**4**) and stephensiolide F (**5**) by comparison with spectroscopic data from available literature (Figure 3) [19,20]. The proposed structures of the isolated molecules can be used to propagate the annotation to other connected MS/MS spectra. By comparison with literature data, the non-isolated molecules could be identified as stephensiolides A, B and E [19,20] (Figure 2B). The absolute configuration of these identified and isolated compounds could not be confirmed.

The 5 isolated compounds from the *Lecanicillium* sp. strain were tested on MRSA. Compound **1** showed a strong activity against MRSA with MIC ≤ 4 µg/mL, compound **3** exhibited activity with MIC ≤ 16 µg/mL, compounds **2** and **5** were found to be less active with MIC ≤ 32 µg/mL, and compound **4** showed a moderate MIC of greater than 128 µg/mL (Table 1). 

## 3. Discussion

It is the first time that some cultivable endophytes have been isolated from latex of the plant *Sandwithia guyanensis* even though the Euphorbiaceae latex is known to be a toxic environment for organisms. The isolated endophytic strains showed antimicrobial effects with MIC values on MRSA, as low as 16 µg/mL. 

Based on the activity observed on antimicrobial assay and using a Molecular Networking-based approach for de-replication, one strain from latex extract, i.e., BSNB-SG3.7, was investigated. BSNB-SG3.7 strain was identified as *Lecanicillium* sp., a fungus in the order Hypocreales, described as anamorphic Cordycipitaceae. Several species of this genus are known as entomopathogenic fungi and some have been developed as commercial biopesticides. Nonetheless, this genus was also described once to be associated with plants. Some isolates of this genus have been shown to possess activity against phytoparasitic nematodes or fungi [21]. 

Many structurally diverse secondary metabolites have been isolated from *Lecanicillium* genus, including indolosesquiterpenoids, phenopicolinic acid analogues, pregnanes, diterpenoids and spiciferone analogues [21,22,23]. In our study, the molecular networks led to the isolation and characterisation of five stephensiolides, as major compounds from our *Lecanicillium* sp. strain extract. Hence, it is for the first time that these compounds are isolated from a fungus. They had been described so far only once, from a bacterium associated with mosquitoes’ metabolites from the *Serretia* genus [19,20]. This is the reason for the absence of these uncommon compounds in the available MS/MS databases.

The diversity in the structure of the isolated stephensiolides resulted from the length of the acyl chain, the presence or absence of a double bond in this chain and also by the presence of isoleucine or valine at the fifth amino acid position. Certain minor products were also observed through MNs, but possibly not discovered yet due to their small amount in our extract.

There are numerous literature reports describing the isolation, characterisation and antimicrobial activities of cyclic lipodepsipeptides [24]. Stephensiolides are cyclised by a lactone bridge between N-terminal hydroxyl group of Thr and C-terminal Ile or Val. Their structures are close to Fusaricidins, that are natural antimicrobial compounds possessing six amino acid residues and a 15-guanidino-3-hydroxypentadecanoic acid attached via amide bond of threonine. However, it was demonstrated for the first time that stephensiolides have an antibacterial activity against gram-positive bacteria, i.e., *S. aureus*. The most active stephensiolides were those having a fatty chain of 11 carbons (stephensiolide I (**1**) and stephensiolide G (**3**)) with MIC on MRSA of 4 and 16 µg/mL, respectively. These two compounds are the most abundant in the extract with respect to the size of the nodes related to the peak area (Figure 2B) and explain the overall activity observed for the crude extract of this strain. 

## 4. Materials and Methods 

### 4.1. Isolation of Natural Substances, Materials and Endophytes

Leaves, wood and latex of two specimens of *S. guyanensis* were collected in October 2018 in St Elie (The Access and Benefit-Sharing Clearing-House (ABSCH) have been obtained and are referenced under the number ABSCH-IRCC-FR-245916-1). Voucher specimens were deposited in Cayenne Herbarium. Endophytes were isolated in less than 24 h after sampling. The isolation method and identification of the endophytic microorganisms were performed according to Casella et al. [21]. After collection, the plant material was washed with sterile water and surface sterilised by sequential immersion in 70% aqueous ethanol (3 min), followed by 5% aqueous sodium hypochlorite (5 min) and finally in 70% aqueous ethanol (1 min). Leaves were cut into small pieces (1–0.5 cm^2^) and placed on Potato Dextrose Agar medium (PDA, Fluka Analytical, Darmstadt, Germany) in Petri dishes (4–5 parts per Petri dish). The latex was collected from fresh plants cut with a sterile cotton bud (sampling kit Probact, Lancashire, UK). Within 24 h, the tip was placed under Microbial safety post in sterile distilled water (2 mL). The water was then distributed on Petri dishes containing PDA (1 mL per dish). The tip where the latex was collected was also directly used to inoculate latex-associated microbes on Petri dishes containing PDA. All Petri dishes were placed at 28 °C. Each individual hyphal tip of emerging fungi or bacterial colony was removed and placed on a sterile PDA culture medium in 10 cm Petri dishes. The leaf fragments were cultured for a maximum of 1 month. All the isolated endophytic strains have been deposited in the ICSN/CNRS Strain library. The strains are maintained in triplicate in 2 mL Eppendorf tubes containing 1 mL solution of glycerol and water (1:1) at −80 °C.

### 4.2. Culture and Extraction of Endophytic Strains

#### 4.2.1. Culture of Endophytic Strains

The stored endophytic microorganisms isolated from *Sandwithia guyanensis* were defrosted and cultivated on a Petri dish containing solid PDA medium and placed at 28 °C. Strains were transplanted and cultivated on 10 Petri dishes for Bacteria and 4 for Fungi. The fungal strains were cultivated for 14 days and bacterial strains for 7 days [25].

#### 4.2.2. Extraction of Endophytic Strains

The endophytic microorganism colonies on the culture medium were cut into small pieces and macerated with ethyl acetate at room temperature on a rotary shaker (70 rpm) for 24 h. The EtOAc extracts were filtered and washed with distilled water in a separating funnel. The organic solvent was collected, dried with anhydrous sodium sulphate and evaporated to dryness under reduced pressure to yield the crude extracts. These crude extracts were stored at 4 °C for further investigation.

#### 4.2.3. Identification of Endophytes

BSNB-SG3.7 strain was submitted for amplification of nuclear ribosomal internal transcribed spacer region ITS1, allowing identification by comparison with NCBI sequence. Briefly, ITS1 sequence was compared with those for the isolate strains present in the NCBI database. Sequences producing significant alignments were used to construct a phylogenetic tree by using the neighbour-joining method with NCBI’s algorithm (default settings). The sequence was found to be registered in the NCBI GenBank database with the accession number MN514023 and identified as *Lecanicillium* sp.

### 4.3. General Experimental Procedures

Nuclear Magnetic Resonance (NMR) spectra were recorded on a Bruker 500 MHz spectrometer equipped with a 1 mm inverse detection probe. ESI-HRMS measurements were performed using a Waters Acquity UHPLC system with column bypass coupled to a Waters Micromass LCT premier Time-of-Flight mass spectrometer (Waters, Milford, MA, USA) equipped with an electrospray interface (ESI). Analytical and preparative HPLCs were conducted with a Gilson system equipped with a 322 pumping device, a GX-271 fraction collector, a 171 diode array detector, and a prepELSII electrospray nebuliser detector. Columns used for these experiments included a Phenomenex Luna PFP (2) 5 μm 4.6 × 250 mm analytical column and Phenomenex Luna PFP (2) 5 μm 21.2 × 250 mm preparative column. The flow rate was set to 1 or 21 mL/min, respectively, using an isocratic elution of CH₃CN-H_2_O (60:40). All solvents were HPLC grade, diluted with 0.1 % trifluoroacetic acid. 

Ultra-High-Performance Liquid Chromatography-High-Resolution Mass Spectrometry (UHPLC-HRMS/MS) analyses were achieved by coupling an Agilent 1260 Prime LC system to a hybrid quadrupole time of flight mass spectrometer Agilent 6540 (Agilent Technologies, Massy, France) equipped with an ESI dual source, operating in positive ion mode. Elution was conducted using Accucore C_18_ column (2.1×100 mm; 2.6 μm, ThermoScientific) with water (A) and acetonitrile (B) as mobile phases, following a gradient 5% –100% B in 20 min, then maintaining 100% B for 4 min at a flow rate of 400 μL.min^−1^. The column oven was set at 30 °C and the injection volume at 5 μL. ESI source parameters were set as follows: capillary temperature at 325 °C, source voltage at 3500 V, sheath gas flow rate at 7 L.min^−1^, nebuliser pressure at 30 psi, drying gas flow rate at 10 L.min^−1^, drying gas temperature at 350 °C, stealth gas temperature at 350 °C, skimmer voltage at 45 V, fragmentor voltage at 150 V and nozzle voltage at 500 V. MS scans were operated in full scan mode from *m/z* 100 to 1000 (100 ms scan time), with a mass resolution of 20,000 at *m/z* 922. MS^1^ scan was followed by MS^2^ scans of the five most intense ions above an absolute threshold of 3000 counts. Selected parent ions were fragmented at a fixed collision energy value of 35 eV and an isolation window of 1.3 amu. Calibration solution, containing two internal reference masses (purine, C_5_H_4_N_4_, *m/z* 121.0509, and HP 921 [hexakis (1H,1H,3H tetrafluoropentoxy) phosphazene], C_18_H_18_O_6_N_3_P_3_F_24_, *m/z* 922.0098), routinely led to mass accuracy below 2 ppm. MS data acquisitions were performed using MassHunter Workstation software (Agilent Technologies, Massy, France).

### 4.4. Biological Tests

#### 4.4.1. Antimicrobial Assays

The endophytic crude extracts and pure compounds isolated were tested on the Human pathogenic microorganisms such as *Candida albicans* (ATCC 10213), MRSA (ATCC 33591) and *Trichophyton rubrum* (SNB-TR1), for the screening of their antibacterial and antifungal activities. The test was performed in conformance with reference protocol from the European Committee on Antimicrobial Susceptibility Testing [25]. The MIC value was obtained after 48 h for *C. albicans*, 24 h for *S. aureus* and 5 days for *T. rubrum*. Vancomycin (for bacteria) and Fluconazole (for fungi) were used as positive controls [26,27,28,29].

#### 4.4.2. Cytotoxicity Assays

The most active crude extracts were tested in triplicate at the concentration of 10 µg/mL and 1 µg/mL on MRC5 cell line (ATCC CCL-171, Human Lung Fibroblast Cells), following the procedure described by Tempête et al. [30]. 

### 4.5. Molecular Networking Analysis

The conversion of MassHunter.d files to.mzXML files was achieved by using MSConvert, a tool included in ProteoWizard package. The converted files were processed by MZmine2 version 2.38 following previously described procedure [31]. The mass detection was realised keeping the noise level at 20000 for all data sets. The ADAP chromatogram builder was completed using a minimum group size of 3 scans, a group intensity threshold of 20000, a minimum highest intensity of 20000, and an *m/z* tolerance of 0.05 or 25 ppm. The ADAP wavelet deconvolution algorithm was configured using the following parameters: S/N threshold of 10, minimum feature height of 20000, coefficient/area threshold of 10, peak duration range of 0.01 of 0.01–1.2 min and RT wavelet range of 0.01–0.07. The *m/z* range for MS^2^ scan pairing was 0.02 Da and the RT range was 0.3 min. Deisotoping was performed using the isotopic peak grouper algorithm with an *m/z* tolerance of 0.05 (or 25 ppm) and RT tolerance of 0.2 min. The peak alignment algorithm was used with the following settings: *m/z* tolerance of 0.003 *m/z* or 10 ppm, weight for *m/z* of 1, retention time tolerance of 0.2 min and weight for RT of 1. The data processing was adapted for each chromatogram. 

The processed data were exported to a.mgf file containing the list of MS/MS spectra, and a.csv file containing information about RT, *m/z*, peak area and identities for the further generation of Molecular Networking on MetGem version 1.2.0 RC2 [17,18]. 

### 4.6. Isolation of Molecules from BSNB-SG3.7 Strain

The total crude extract of BSNB-SG3.7 (62 mg) was first purified by SPE using Waters Sep-Pak Vac C_18_ Cartridges. Recovered eluate was then evaporated to yield 50 mg extract and subjected to a preparative HPLC using an isocratic elution described above, to yield stephensiolide I (**1**) (4.3 mg, RT 26.1 min), stephensiolide D (**2**) (1.3 mg, RT 11.6 min), stephensiolide G (**3**) (1.5 mg, RT 18.5 min), stephensiolide C (**4**) (0.6 mg, RT 8.7 min) and stephensiolide F (**5**) (1.9 mg, RT 17.2 min). The time was 35 min for each analysis. 

Identity of the compounds were determined by comparing with available literature, see SI (Appendix A and Appendix A)

## 5. Conclusions

To our knowledge, our work reports the first chemical study on latex endophytes. First, it was demonstrated that latex endophytes can biosynthesise antimicrobial compounds. Further, MN method allowed us to easily identify target compounds responsible for the observed biological activity on MRSA. Indeed, MN led us to isolate antimicrobial molecules from one active strain isolated from *S. guyanensis* latex, *Lecanicillium* sp. (BSNB-SG3.7).

Our study highlighted that the *Lecanicillium* sp. can produce diverse stephensiolides with antimicrobial activity on methicillin-resistant *Staphylococcus aureus*, which forms a pioneer documentation of antibacterial activity in these compounds. Also, 5 stephensiolides were isolated from fungi for the first time. Furthermore, the MN-based approach revealed several close analogues of stephensiolides and other known stephensiolides in the *Lecanicillium* sp. Extract. A large-scale experimentation would be required to isolate and identify the minor compounds observed in MNs and also the original compounds produced by other active strains. 

Stephensiolides are known to aid bacterial swarming motility and have been reported to have a role in interspecific interactions that contribute to transmission of viruses [18] The role of these compounds in a host plant require further detailed investigation. To our knowledge, no *Lecanicillium* spp. has been developed for control of phytopathogens or human pathogens to date.

## Figures and Tables

**Figure 1 plants-09-00047-f001:**
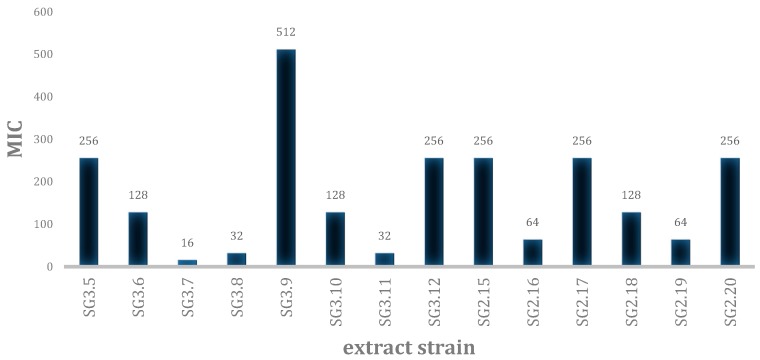
Minimal Inhibitory Concentration (MIC) of the extract strains isolated from *S. guyanensis* latex on methicillin-resistant *Staphylococcus aureus* (MRSA).

**Figure 2 plants-09-00047-f002:**
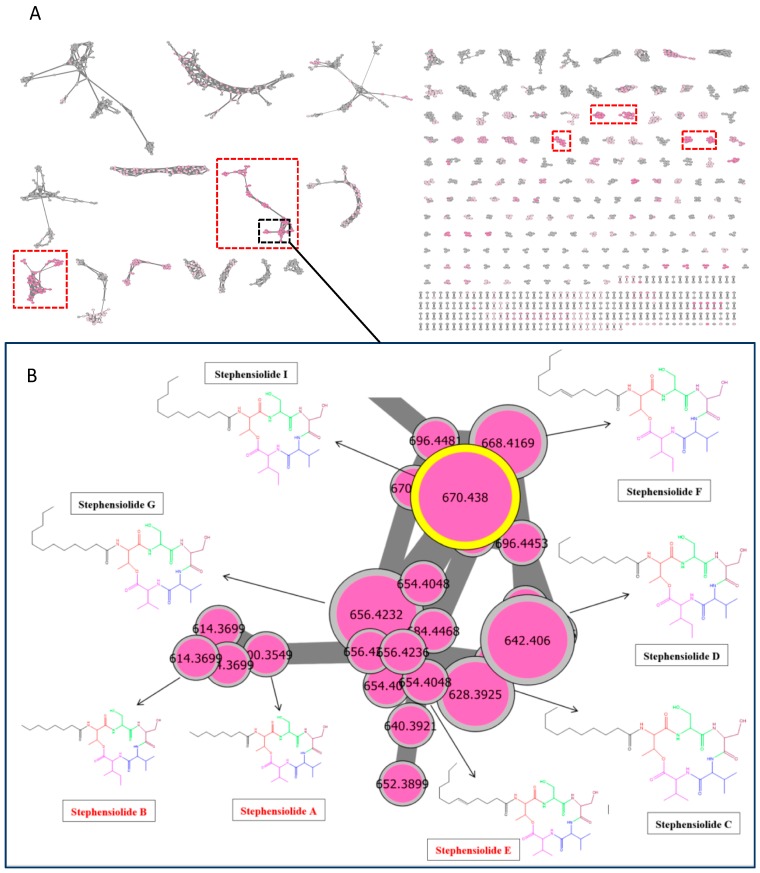
(**A**) Global Molecular Networks of 42 strain extracts isolated from *S. guyanensis*. The clusters from active extracts on MRSA are shown in pink. Clusters from BSNB-SG3.7 strain are surrounded by red. Cluster of analogues of stephensiolides from *Lecanicillium* sp. (BSNB-SG3.7) extract strain is surrounded in black. (**B**) Isolated compounds of BSNB-SG3.7 are shown in black and the predicted compounds are in red.

**Figure 3 plants-09-00047-f003:**
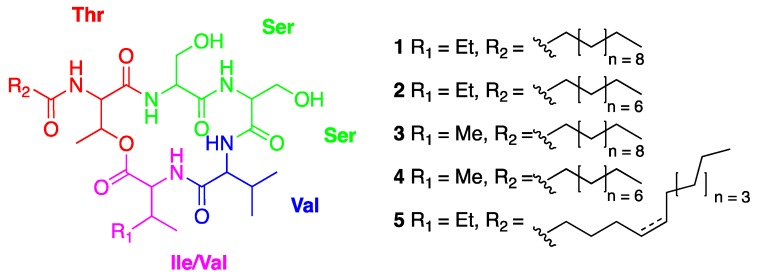
Structures of the isolated compounds **1** to **5**.

**Table 1 plants-09-00047-t001:** Antimicrobial activities of compounds 1–5.

Compound	MIC on MRSA (µg/mL)
**1**	4
**2**	32
**3**	16
**4**	128
**5**	32
**BSNB-SG3.7 extract**	16
**Positive Control**	0.6

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
