# Peer review of "Identification of Antimicrobial Compounds from Sandwithia guyanensis-Associated Endophyte Using Molecular Network Approach"

_plants, 2019, doi:10.3390/plants9010047_

Round 1

Reviewer 1 Report

This manuscript is about the identification of antimicrobial compounds from Sandwithia guyanensis-associated endophyte using molecular network approach. The scope of the study is clear with possible application in therapeutics of bacteria and fungi infections.

The manuscript should be edited by a native speaker. There are several major grammar and syntax errors.

Specific comments.

Lines: 18 and elsewhere. The family name of plants should be italicized.

Lines: 31-33. The authors mentioned about antibiotics resistance among Gram-negative bacteria. However manuscript is focus on Gram-positive bacteria (S. aureus) and fungi (C. albicans, T. rubrum). In my opinion, it should be corrected.

Lines: 69-76. The authors mainly focused on antimicrobial activity of 42 cultivable endophytic extracts against human pathogens. The ATCC reference strains were used in this study. Both, reference and clinical strains have vary virulence, which was proved before by many scientists. Hence, it should be corrected.

Lines: 45 and elsewhere. First person report should be avoided. Third person is more appropriate.

Lines: 208-212. Author contributions (full name) should be avoided. This should be in Acknowledgments section.

Page 7 - Culture and extraction of endophytic strains (4.2). The authors should consider in future an identification of all strains isolated from Sandwithia guanensis before extraction with ethyl acetate. BSNB-SG3.7 should be more precisely identified. According to my best knowledge there are 21 species of Leacanicillium genus.

Lines: 280-282. The authors performed antimicrobial assays using the European Committee on Antimicrobial Susceptibility Testing (EUCAST) protocol. However, there is no reference to this protocol.

Table S1: It should be corrected: MIC of C. albicans etc. MIC SARM should be corrected.

Generally, this is an interesting paper that could be published after major edits.

Author Response

This manuscript is about the identification of antimicrobial compounds from Sandwithia guyanensis-associated endophyte using molecular network approach. The scope of the study is clear with possible application in therapeutics of bacteria and fungi infections.

The manuscript should be edited by a native speaker. There are several major grammar and syntax errors.

Thanks to the reviewers for having accepted to evaluate our publication. Please see our answers attached to your questions in blue in the document below and highlighted in blue in mansucrit

The document was reviewed by Elsevier language editing services, find in attached document the certificate.

Specific comments.

Lines: 18 and elsewhere. The family name of plants should be italicized.

We have italicized the name of the plant as what is requested in Plants nomenclature. In botany, until now family names are not to be italicized. There is some confusion as to how family names should be written. In American usage the family name is not usually italicized, e.g. Euphorbiaceae, however the most recent edition of the International Code of Nomenclature for algae, fungi, and plants (which is the official authority on plant names) recommends that all plant names be in a different font from the rest of the text. The Royal Horticultural Society (U.K.) recommends that family names be italicized. Plant labels in botanical gardens usually have the family name in capital letters,  e. g. EUPHORBIACEAE. As this nomenclature mode is not clearly defined, we chose to not modify the manuscript.

Lines: 31-33. The authors mentioned about antibiotics resistance among Gram-negative bacteria. However manuscript is focus on Gram-positive bacteria (S. aureus) and fungi (C. albicans, T. rubrum). In my opinion, it should be corrected.

Thank you for this remark, we have generalized antibiotic resistance to that of gram negative bacteria. We have revised the manuscript as indicated.

Lines: 69-76. The authors mainly focused on antimicrobial activity of 42 cultivable endophytic extracts against human pathogens. The ATCC reference strains were used in this study. Both, reference and clinical strains have vary virulence, which was proved before by many scientists. Hence, it should be corrected.

As indicated we have reported that we have used ATCC strains for MRSA and C. albicans but for T. rubrum it is a clinical strain.

Lines: 45 and elsewhere. First person report should be avoided. Third person is more appropriate.

Done

Lines: 208-212. Author contributions (full name) should be avoided. This should be in Acknowledgments section.

Done

Page 7 - Culture and extraction of endophytic strains (4.2). The authors should consider in future an identification of all strains isolated from Sandwithia guanensis before extraction with ethyl acetate. BSNB-SG3.7 should be more precisely identified. According to my best knowledge there are 21 species of Leacanicillium genus.

With ITS sequencing (the latter is deposited on Genbank) we do not have the possibility to formally determine the species, it seems that our strain is close to the species "aphanocladii" but we prefer to remain prudent. The phylogenetic tree obtained by comparison with the sequencing data of other species of the same genus (http://libanswers.nybg.org/faq/223266) was send.

Lines: 280-282. The authors performed antimicrobial assays using the European Committee on Antimicrobial Susceptibility Testing (EUCAST) protocol. However, there is no reference to this protocol.

This reference was added

Table S1: It should be corrected: MIC of C. albicans etc. MIC SARM should be corrected.

Done

Generally, this is an interesting paper that could be published after major edits.

Reviewer 2 Report

This paper investigated the compounds isolated from plant sandwithia guyanensiswith antimicrobial activity on MRSA and other microorganisms. It is a topic of interest to the researchers in the related areas. But the paper needs some improvement before acceptance for publication. My detailed comments are as follows:

In the abstract, the MIC value should be described of two potent active compounds (1 and 3) In page 5, line 161, compound 1 should also be included as stephensiolide I. In page 5, line 165 and page 6, line 204, Figure 2C was mentioned, but there is no Figure 2C in the manuscript, please add it.

Author Response

This paper investigated the compounds isolated from plant sandwithia guyanensiswith antimicrobial activity on MRSA and other microorganisms. It is a topic of interest to the researchers in the related areas. But the paper needs some improvement before acceptance for publication. My detailed comments are as follows:

We thank you for taking the time to evaluate our manuscript and for pointing out some errors

In the abstract, the MIC value should be described of two potent active compounds (1 and 3) In page 5, line 161, compound 1 should also be included as stephensiolide I. In page 5, line 165 and page 6, line 204, Figure 2C was mentioned, but there is no Figure 2C in the manuscript, please add it.

We thank you for this remark and, as proposed, we have included the names of the molecules in the text. For the Figure 2B, it’s a mistake it was figure 2B that is quoted. Thank you for your corrections.

Round 2

Reviewer 1 Report

Table S1: I didn't see change in Table S1. You should change SARM to MRSA (it mean: methicillin-resistant Staphylococcus aureus). Include the legend of Table S1 plus acronyms (MRSA, MIC).

Line: 286. The authors should add EUCAST recommendation in Reference section, instead Materials and methods section.

Author Response

Thank you for your corrections, they have been implemented as requested.